# Impact of Immunotherapy on CD4 T Cell Phenotypes and Function in Cancer

**DOI:** 10.3390/vaccines9050454

**Published:** 2021-05-04

**Authors:** Margaux Saillard, Mara Cenerenti, Pedro Romero, Camilla Jandus

**Affiliations:** 1Department of Oncology, University of Lausanne, 1006 Epalinges, Switzerland; margaux.saillard@unil.ch (M.S.); pedro.romero@hospvd.ch (P.R.); 2Department of Pathology and Immunology, Faculty of Medicine, University of Geneva, 1211 Geneva, Switzerland; mara.cenerenti@unige.ch; 3Ludwig Institute for Cancer Research, Lausanne Branch, 1006 Epalinges, Switzerland

**Keywords:** CD4 T cells, immunotherapy, antigen-specific, cancer, immune monitoring

## Abstract

Immunotherapy has become a standard treatment in many cancers and it is based on three main therapeutic axes: immune checkpoint blockade (ICB), vaccination and adoptive cell transfer (ACT). If originally these therapies mainly focused on exploiting CD8 T cells given their role in the direct elimination of tumor cells, increasing evidence highlights the crucial role CD4 T cells play in the antitumor immune response. Indeed, these cells can profoundly modulate the tumor microenvironment (TME) by secreting different types of cytokine or by directly eliminating cancer cells. In this review, we describe how different CD4 T cell subsets can contribute to tumor immune responses during immunotherapy and the novel high-throughput immune monitoring tools that are expected to facilitate the study of CD4 T cells, at antigen-specific and single cell level, thus accelerating bench-to-bed translational research in cancer.

## 1. Summary

Immunotherapy has become a standard treatment in many cancers and it is based on three main therapeutic axes: immune checkpoint blockade (ICB), vaccination and adoptive cell transfer (ACT). If originally these therapies mainly focused on exploiting CD8 T cells given their role in the direct elimination of tumor cells, increasing evidence highlights the crucial role CD4 T cells play in the antitumor immune response. Indeed, these cells can profoundly modulate the tumor microenvironment (TME) by secreting different types of cytokine or by directly eliminating cancer cells. In this review, we describe how different CD4 T cell subsets can contribute to tumor immune responses during immunotherapy. We emphasize the need to better understand the dynamics of tumor antigen-specific CD4 T cell responses, as opposed to bulk studies, to dissect the contribution of tumor-specific versus bystander T cell responses. Particularly, we review current knowledge of CD4 T cell fate dimensions at antigen-specific level, longitudinally in clinical and preclinical studies. These observations, in combination with the development of novel high-throughput immune monitoring tools, are expected to facilitate the study of CD4 T cells, at antigen-specific and single cell level, thus accelerating bench-to-bed translational research in cancer and biomarker discovery.

## 2. CD4 T Cell Functional Polarization and Tumor Immunity

CD4 T cells residing in the TME can acquire a defined functional, highly plastic cell fate characterized by their ability to selectively produce cytokines in response to various cellular signals and their master transcription factor expression [1]. A growing number of different subsets, including Th1, Th2, Th9, Th17, Th22, Tfh, Th*, Treg and Th-CTX have been described (Figure 1) and were shown to distinctly affect the prognosis of cancer patients. On the one hand, CD4 T cell subsets have been firmly associated with beneficial effects on antitumor immunity. Th1 cells are linked in several cancer types with the establishment of efficient antitumor responses [2,3]. Indeed, they are associated with a good prognosis in many cancers such as non-small cell lung cancer (NSCLC) [4] or colorectal cancer (CRC) [5] where the frequency of tumor infiltrating Th1 polarized CD4 T cells is directly linked with a good clinical outcome. It has also been reported that Th9 cells have the ability to control and to suppress tumor growth [6]. For instance, it has been shown that high Th9 cell numbers in patients with NSCLC are associated with a better prognosis [7] and additional studies reported the tumor growth control capacity of Th9 cells in melanoma [8,9,10] and in colon carcinoma [11], in an IL-9 dependent manner. Th* (alternatively called Th1/Th17) cells have been carefully characterized in mycobacterial infections, while their role is less well defined in cancer [12]. Still some evidence highlights their involvement in tumor regression in preclinical models [13]. Moreover, Tfh cells have been positively associated with long-term survival of patients with breast cancer, through CXCL13 secretion [14], with still some studies showing a correlation between high percentage of Tfh cells and advanced-stage disease, such as in NSCLC [15] or gastric cancer [16]. Finally, fully cytotoxic Th-CTX cells [17] are gaining importance as effectors in cancer, particularly in the context of MHC class I loss. Indeed, we recently identified CD4 T cells with cytotoxic phenotypes involved in the direct elimination of tumor cells in different human cancer types [18]. Overall, these observations highlight the potential of targeting CD4 T cells to increase the potency and antitumor activity of immunotherapy. On the other hand, however, functional specialization of CD4 T cells towards other polarities may have the opposite effect and promote tumor growth and dissemination. For instance, Th2-type inflammation at the tumor site facilitates carcinogenesis and tumor progression in cervical carcinoma by creating an immunosuppressive environment [19]. Moreover, various experimental models demonstrate that Th17 cells act differently in tumor immunity, having both anti- and protumorigenic roles. For instance, some studies show that Th17 cells increase tumor progression by activating angiogenesis and immunosuppressive activities [20], but they also display antitumor roles by increasing natural killer (NK) and cytotoxic CD8 T cell (CTL) triggering and by recruiting neutrophils, NK, CD4 and CD8 T cells into the TME. Moreover, Amicarella et al. showed this dual role of Th17 cells in CRC, where CRC-derived Th17 cells triggered the release of protumorigenic factors by the tumor, while favoring the recruitment of beneficial neutrophils [21]. Th22 cells also promote tumorigenesis in some cancers such as colon [22] and gastric cancer [23]. Indeed, it was shown that Th22 cells increased as tumor stage advanced and their frequency was predictive of reduced overall survival. Finally, Treg cells are a highly immune-suppressive fraction of CD4 T cells and are mainly associated with unfavorable prognosis [24]. Yet, in some cancers, the presence of high densities of Treg cells correlate with a good prognosis, such as in CRC [25,26]. Overall, during natural tumor immunity Th and Treg cells play a key role in orchestrating both anti- and procancer responses. Tumors commonly escape from the elimination by the immune system by using multiple strategies, notably the active suppression and the modulation of effector immune cells. A better understanding on how these cells interact with cancer is necessary and essential to develop immunotherapies that favor a Th/Th-CTX-tumor eliminating rather than a Th/Treg-tumor promoting balance.

## 3. CD4 T Cells in Immune Checkpoint Blockade (ICB) Therapy

Immune checkpoint blockade in cancer disrupts immune regulatory negative circuits to unleash full effector functions of immune cells. Monoclonal antibodies targeting CTLA-4, PD-1 and PD-L1 are agents that have already had a major impact in cancer immunotherapy [27] (Table 1). They were shown to regulate the abundance and functions of antigen experienced CD4 T cells, such as restoring the antitumor properties of Th subsets and depleting Treg cells [28,29]. Mechanistically, the action of CTLA-4 and PD-1 blockers is at least in part distinct. While both unleash exhausted CD8 T cell responses, irrespective of the tumor type, anti-CTLA-4 antibodies additionally modulate the CD4 T cell effector compartment, by expanding an effector ICOS^+^ CD4 T cell subset [30]. Further, using global systemic immune cell profiling by CyTOF and single cell RNAseq in tumor bearing animals undergoing immunotherapy, a peripheral CD4 T cell cluster was identified as being sufficient to mediate antitumor response. This population was characterized by a Th1 effector memory profile in mice and was also identified in peripheral blood of melanoma patients responding to anti-CTLA-4 antibodies in combination with granulocyte-macrophage colony-stimulating factor (GM-CSF) [31]. By blocking the interaction between PD-1 and its ligand PD-L1 with Nivolumab, Pembrolizumab or Atezolizumab, respectively, among others, it is possible to restore T cell responses which were previously attenuated [32]. Zuazo et al. showed that 70% of lung cancer patients with high baseline percentages of memory CD4 T cells and PD-L1-positive tumors respond to therapy, arguing for the requirement of a pre-existing systemic CD4 T cell immunity for successful clinical response [33]. In a recent report, terminally exhausted CD4 T cells were identified in the TME of several solid tumors. PD-1 blockade restored helper activity of exhausted PD-1^high^, CD39^+^ tumor infiltrating CD4 T cells [34]. In MHC-II-expressing tumors such as classic Hodgkin lymphoma, CD4 T cells were significantly associated with the clinical efficacy of PD-1 blockade [35]. However, it has been proposed that hyper-progression of cancer may occur when PD-1 blockade activates and expands tumor-infiltrating PD-1^+^ Treg cells, ultimately overwhelming tumor-reactive PD-1^+^ effector T cells, as reported in approximately 10% of advanced gastric cancer patients [36]. Despite the superior efficacy of the anti-PD-1 antibodies, there are treatment failures and needs for alternative strategies to be considered for these patients. Bowyer et al. addressed this aspect in a study of second-line treatment with an anti-CTLA-4 antibody (Ipilimumab), after failure of anti-PD-1 therapy in advanced melanoma patients. They showed that Ipilimumab can induce responses in these patients [37]. Indeed, 10% of melanoma patients achieved an objective response to Ipilimumab, and 8% experienced prolonged stable disease (more than 6 months). Unfortunately, the mechanism of action of CTLA-4 is only partially understood. Romano et al. tried to highlight this mechanism in advanced melanoma patients and showed that anti-CTLA-4 effects are mediated by the modulation of Treg cell activity and/or by the Fc portion of the antibody itself [38]. While depletion of Tregs in the TME by antibody dependent cellular cytotoxicity (ADCC) mediated by anti-CTLA-4 antibodies has been clearly demonstrated in mouse models [39], the role of this mechanism in the anti-CTLA-4 effect in humans remains a matter of controversy [40]. There are several other strategies to further improve ICB therapies, by fostering the induction and mobilization of T cell responses. For instance, ICB were used in combination with vaccines, such as by Zaidi et al. who used anti-CTLA-4 antibodies with dendritic cell (DC)-targeted vaccines to promote IL-3^+^ CD4 T cell infiltration into mouse pancreatic cancer [41]. By acting on key regulators of immune tolerance, ICB therapy is associated with immune related adverse events (irAEs) [42]. irAEs in ICB are autoimmune-like manifestations that can affect any organ, most commonly the skin, colon, lungs, liver and thyroid. ICB-related irAEs occurs in 80–90% of treated patients, and increased severity is observed upon combinations of CTLA-4 blockade with anti-PD-1 or anti-PD-L1. Endocrine and rheumatologic toxicities might become chronic [43], arguing for the need to carefully assess treatment combinations to increase clinical benefit by limiting toxicities. Besides PD-1 and CTLA-4, there are also novel emerging checkpoint targets such as the V domain immunoglobulin suppressor of T cell activation (VISTA) which plays a critical role in antitumor immunity. Human VISTA is highly expressed on myeloid cells with reduced expression on CD4 and CD8 T cells. Moreover, VISTA is suppressive on both resting and activated human T cells, making it a potent negative regulator of T-cell function and a good target in cancer treatment [44,45]. A single cell RNA-sequencing study on breast tissues [46] showed a higher level of VISTA expression in the cancer tissue compared to adjacent normal tissue. These results support the immunoregulatory role of VISTA in breast cancer and suggest that targeting VISTA may benefit breast cancer immunotherapy. Lymphocyte activation gene-3 (LAG-3) is also a promising target for cancer immunotherapy. It is a type I transmembrane protein with structural similarities to CD4 [47]. It could be a potential target for combination therapies with PD-1 blockade. Indeed, Woo et al. showed that there is a synergistic cooperation between LAG-3 and PD-1 in limiting tumor growth [48] and, recently, a soluble LAG-3 protein in combination with anti-PD-1 was tested in patients with metastatic melanoma [49]. Another checkpoint target candidate is CCR4, that plays an important role in regulating the immune balance and is highly expressed in Treg cells. Treatment with an anti-CCR4 antibody selectively depleted effector Treg cells and efficiently induced tumor-antigen-specific CD4 and CD8 T cells [50].

Agonistic antibodies targeting costimulatory receptors may provide additional opportunities for immunotherapeutic intervention if immune related adverse events may be maintained within an acceptable safety profile. Costimulatory receptors on T cells that have been targeted for immunotherapy of cancer include OX40, GITR and 4-1BB. Neoadjuvant anti-OX40 therapy in patients with head and neck squamous cell carcinoma led to increases of activated CD4 and CD8 T cells in both blood and tumors [51]. The initial results of therapy with an agonistic anti-GITR antibody indicates that it leads to reduction of regulatory CD4 T cells in both blood and tumors and would need to be combined with other immunotherapies, anti-PD-1 specifically, to attain clinical efficacy [52].

## 4. CD4 T Cells in Cancer Vaccination

### 4.1. Tumor-Antigenic Peptides and Adjuvants

Peptide based vaccines are composed of tumor-antigenic peptides and one or more immune stimulatory adjuvants. Vaccination in cancer patients has initially been focused on targeting CD8 T cells but the role of CD4 T cells is becoming more evident in antitumor vaccinations. After initial exploratory trials with short exact CD8 T cell-defined cytotoxic T cell epitopes, it became clear that long synthetic peptides were needed not only to selectively engage professional antigen cross-presenting cells but also to provide CD4 helper T cell activation [53,54] (Figure 2). A comparison of a panel of adjuvants on antitumor responses upon peptide vaccination in tumor mouse models showed that CPG-ODNs and PolyI:C promoted the expansion of antigen-specific CD8 and CD4 effector T cells compared to QuilA and Imiquimod that decreased the effector T cells [55]. In line with these preclinical observations, it was reported that strong and long-lasting CD4 T cell memory responses of Th1 type, generally associated with antitumor responses, are induced in advanced melanoma patients upon vaccination consisting of the synthetic NY-ESO-1_119–143_ peptide, Montanide and CpG-ODNs as adjuvants [56]. The long synthetic NY-ESO-1_79–108_ peptide combined with the strong immune adjuvant CpG-B was also shown to lead to robust and functional CD8 and NY-ESO-1_79–108_ specific CD4 T cell responses, with type 1 polarity being associated with longer survival [57]. Concomitant antigen-specific CD8 and CD4 T cell responses were also observed upon vaccination with the Melan-A_26–35_ peptide that triggered a switch from Treg to Th1 polarized tumor-specific CD4 T cells in HLA-DQ6-vaccinated melanoma patients [58].

Further, cervical cancer is a disease predominantly induced by human papilloma virus types 16 and 18 (HPV-16, -18). In preclinical models, vaccination with the HPV-16-derived 35 amino-acid long peptide E7_43–77_ and with the DC-activating adjuvant CpG-ODN, lead to the generation of E7-specific CD4 T cells [59]. Immunogenicity of this peptide was then confirmed in a clinical trial in women with HPV-16 positive, high-grade vulvar intraepithelial neoplasia (VIN). After vaccination with a synthetic long-peptide containing nine HPV-16-E6 and four HPV-16-E7 synthetic peptides of 24–34 amino acid length in incomplete Freund’s adjuvant, the researchers observed a regression of positive lesions and a complete regression in 79% and 47% of patients, respectively, 1 year after the last dose of the vaccine [60]. Successive trials further attested the effectiveness of this therapy for HPV-16-induced high-grade VIN patients [61,62]. Beside individual long peptides, also overlapping peptides have been considered and tested in clinical studies. It was shown that vaccination using overlapping synthetic long NY-ESO-1 peptides in cervical cancer induced a consistent NY-ESO-1 specific CD4 T cell response in all patients when Montanide was used as adjuvant compared to PolyI:C [63].

Of note, recent evidence shows that CD4 T cells can efficiently recognize neoantigens. Neoantigens offer a new and interesting target for personalized cancer immunotherapy due to their neo-expression on cancerous tissues, therefore not being considered as self-antigens. In an initial study, neoantigen-reactive CD4 T cells were detected in four out of five melanoma patients analyzed, including subjects who had a clinical response after adoptive T cell therapy [64]. In a successive study, a peptide vaccine that targeted up to 20 predicted personal tumor neoantigens was developed. Vaccine-induced polyfunctional CD4 and CD8 T cells targeted some unique neoantigens used across patients and four out of six vaccinated patients had no recurrence at 25 months after vaccination [65]. More recently, the same group developed a personal vaccine with a long-peptide targeting up to 20 neoantigens per patient, in order to induce and diversify the antitumor T cell response [66]. They observed CD4 and CD8 neoantigen-specific T cell responses, and after assessing gene expression profiles in individual neoantigen-reactive CD4 T cells directly ex-vivo they found an upregulation of genes related to cytotoxicity such as granzyme A and granulysin, suggesting that these vaccine-specific CD4 T cells may be able to kill tumor targets directly, beside their helper functions [67].

### 4.2. RNA-Based Vaccines

RNA vaccines represent a promising alternative to conventional vaccine approaches because of their high potency and their capacity for rapid development. mRNA encoding one or more immunogens of interest is lipid-encapsulated to avoid the degradation by extracellular RNAses and to improve the internalization efficiency [68]. Then, mRNA can be delivered into the host cell cytoplasm where its expression generates translated proteins to be within the membrane, secreted or intracellularly located [69,70]. mRNA vaccine platforms against several types of cancer have demonstrated encouraging results in both animal models and humans [71]. The mRNA-based poly-neoepitope approach to mobilize immunity against a spectrum of melanoma mutations [72] induced antitumor activity after targeting individual cancer mutations by both CD4 and CD8 T cells (Figure 2). Rejection of CT26 tumors in mice, mediated by local radiotherapy, is further augmented in a CD8 T cell-dependent manner by an RNA-LPX vaccine that encodes CD4 T cell-recognized neoantigens [73]. More recently, the group of Rosenberg [74] developed a novel mRNA vaccine encoding defined neoantigens, mutations in driver genes and HLA-I predicted epitopes in gastro-intestinal cancer patients. They could detect both CD4 and CD8 neoantigen-specific T cells but interestingly, the vaccine elicited mainly CD4 and not CD8 T cell specific responses. This observation might be explained by the high expression levels of the vaccine mRNA that gives rise to peptides that can be presented on antigen-presenting cells (APCs) and stimulate CD4 T cells showing, once again, the importance of these lymphocytes in antitumor responses. The mRNA vaccines delivered in lipid nanoparticles intravenously can also elicit strong CD4 and CD8 specific T cell responses against non-mutated tumor associated antigens (TAAs) alone or in combination with anti-PD-1 blockade. A high objective response rate was observed with this combination in advanced metastatic melanoma [75]. The formation of complexes with the cationic compound protamine, instead of lipids, has also been successfully used to deliver mRNA encoding tumor specific antigens in phase I clinical studies in combination with local irradiation [76].

### 4.3. DNA- and Recombinant Vector-Based Cancer Vaccines

DNA-based vaccines involve the use of a DNA plasmid which encodes for a tumor antigen to target cancer. They are cost-efficient, stable, and safe in handling (Figure 2). The effectiveness of a DNA vaccine containing two plasmids encoding a fusion protein of the HPV-16 E6 and E7 viral sequences was tested in a phase I clinical trial in HPV-associated oropharyngeal squamous cell carcinoma patients, previously treated by radiotherapy [77]. An enhancing of the specific immunity to virus-derived TAAs in patients was observed. Similarly, patients with cervical intraepithelial neoplasia (CIN) were treated with synthetic plasmids targeting HPV-16 and HPV-18 E6 and E7 proteins. Both histopathological regression and viral clearance after therapeutic vaccination compared with placebo were observed [78]. Further, a vaccine encoding a fusion protein of the HPV-16 E6 and E7 viral sequences, combined with the extracellular domain of Flt3L to promote antigen presentation, suggested significant antitumor and viral clearance efficacy to treat CIN3 patients in a prospective randomized phase II clinical trial [79]. DNA vaccines encoding the prostatic acid phosphatase (PAP) have also been used in metastatic prostate cancer patients. These were tested in a prime-boost vaccination study in combination with the well-known Sipuleucel-T vaccine (see also the Section DC-based vaccination) [80]. An augmentation and diversification of the type of immunity elicited with vaccination, in terms of T cell responses, were observed. However, the phase II clinical trial failed to demonstrate an overall increase in 2-year survival in patients with castration-sensitive prostate cancer [81]. A fusion vaccine consisting of the HLA-A2-binding peptide CAP-1 from the carcioembryogenic antigen (CEA) and the immunogenic domain from fragment C of the tetanus toxin was tested in an exploratory Phase I/II study in patients with CEA-expressing tumors. The rational of this approach is to elicit non-tolerized CD4 T cell responses to the tetanus antigen that would help stimulate CEA-specific cytolytic CD8 T cell immunity. Immunization induced CEA-specific CD8 T cell responses were detectable in the peripheral blood of post-vaccine samples. Further, a decrease in CEA was observed in advanced disease patients experiencing a better overall survival upon vaccination [82]. Recombinant vector-based cancer vaccines are showing promising results. In contrast to naked DNA vaccines, the vector-based vaccines do not require electroporation and may induce generally stronger immune responses. However, immunity directed against the vector backbones may outcompete and dampen responses elicited by the tumor antigen(s) inserts. *Listeria* strains encoding the HPV-16 E7 protein with two different expression systems, Lm-E7 and Lm-LLO-E7, have been tested. Both secrete the E7 tumor antigen and are able to induce an antitumor response. However, Lm-LLO-E7, which secretes a fusion protein consisting of a truncated listeriolysin-O joined at the C-terminus to E7, can cure the majority of treated tumor bearing mice, while the Lm-E7, which secretes the recombinant protein E7, had little impact on tumor growth [83]. A randomized phase 2 study of Lm-LLO-E7 with or without cisplatin in patients with recurrent/refractory cervical cancer showed promising safety and efficacy results [84]. Cappuccini et al. used recombinant adenovirus-based vaccines in a first-in-human study to evaluate 5T4 viral vectored vaccination in early-stage prostate cancer patients [85]. They observed T cell responses in the circulation and in the prostate gland and are currently moving ahead to a Phase I/II clinical trial. Finally, recombinant vaccinia-based cancer vaccines also showed promising results. Indeed, Tosch et al. developed a therapeutic viral vaccine encoding human Mucin 1, which is a TAA, and interleukin-2. In total, 78 NSCLC patients, also treated by chemotherapy, received a vaccine injection. Longer overall survival at 35-month was observed in patients treated with the vaccine in comparison to patients treated with standard chemotherapy only. This survival improvement is correlated with an enhancing of T cell responses against Mucin 1 and additional TAAs, attesting for epitope spreading enriching for diversity of the antitumor response [86].

Although these approaches are mainly focused on CD8 T cell targeting, similar work can be considered to drive CD4 T cell responses.

### 4.4. Protein-Based Vaccines

Protein-based vaccines have the advantage of potentially inducing the full range of epitopes recognized by both CD4 and CD8 T cells (Figure 2). In addition, protein vaccination leads to presentation of epitopes for various HLA alleles and so this type of vaccine does not need to consider the HLA restriction of the patients. To date, successful clinical studies using MAGE-A3 protein as a vaccine have been reported [87,88]. Atanackovic and colleagues showed a strong peptide-specific Th1 type CD4 T cell response using a MAGE-A3 protein vaccine in patients with lung cancer [89]. They demonstrated that protein vaccination can induce CD4 T cell responses that correlate with antibody production. Other clinical trials showed the efficacy of using a NY-ESO-1 protein-based vaccine optimized by the addition of the adjuvant PolyI:C in melanoma patients. In particular this vaccine was able to elicit a strong NY-ESO-1 specific CD4 Th1 and humoral response, while in combination with another adjuvant, Montanide, the vaccine induced a specific CD8 T cell response and a shift of the CD4 T cell polarization towards Th2 phenotypes. Both T cell subsets produced IFNγ, TNF and IL-2 in response to the antigen. However, the pivotal phase III trials in melanoma [90] and lung cancer [91] unfortunately failed to show clinical efficacy. Further, clinical trials in lymphoma patients were exploiting recombinant idiotype vaccines. Idiotypes are the variable regions of the immunoglobulin that binds to the antigen and since they are expressed in a unique pattern on a given malignant B cell, they can serve as a tumor-specific antigen to elicit antitumor CD4 and CD8 T cell responses: the results of different trials using this strategy in immunotherapy has been nicely reviewed by Kwak and colleagues [92].

## 5. CD4 T Cells in the Context of DC-Based Vaccination

DCs are tissue resident and circulating cells also called “immunological sentinels” because they efficiently recognize and internalize, process and then display peptides of pathogens on their surface [93,94,95], including those that are commonly expressed in the TME by the cancer cells themselves, known as TAAs. In this way, DCs can activate TAA-specific T cells to generate an efficient antitumor response [96,97] mediated by both CD4 and CD8 T cells [98]. CD4 T cells are preferentially primed by cDC2s, while CD8 T cells by cross-presenting cDC1s, that in turn are “licensed” by the help of CD4 T cells [99]. Consequently, DCs are considered a good vehicle for T-cell stimulating antitumor vaccines (Figure 3). In 2010 the FDA approved the first anticancer DC-based vaccine, named Sipuleucel-T, for treatment of patients with hormone refractory prostate cancer. This vaccine is composed of autologous DCs displaying a fusion recombinant protein which is made of the whole human PAP, an antigen expressed in prostate cancer but not in nonprostatic tissue, fused with an adjuvant GM-CSF. The authors demonstrated that when the vaccine is infused into patients, DCs induce an immune response against the PAP by activating both CD4 and CD8, while GM-CSF sustain the maturation of the DCs, providing a survival advantage in hormone refractory prostate cancer patients [100].

Initially, only MHC class I-restricted peptides were part of DC-based vaccines leading to the exclusive activation of CD8 T cells. Increasing knowledge shows the importance to elicit also CD4 T cells, in particular in cases where tumor cells downregulate MHC class I and mainly express MHC class II molecules, like in melanoma, lung cancer, breast and osteosarcomas [101]. In that regard, it has been reported that the targeting of CD4 T cells with DCs pulsed with both MHC class II and MHC class I restricted epitopes of the TAA MAGE-A3 enhances the efficacy of the response in vaccinated melanoma patients, although the frequency of antigen-specific CD4 T cell in the patients remained low [102]. Importantly, MAGE-A3-specific CD4 T cells acquired a typical Th1 phenotype and secreted IFNγ and TNF, but not IL-4 and IL-10. By sequencing the TCRs the authors revealed a diverse and polyclonal immune response against the MAGE-A3_243–258_ epitope. Further, Aarntzen and colleagues demonstrated improved clinical outcome of melanoma patients receiving a vaccine consisting of DCs pulsed with MHC class I and II restricted epitopes compared to DCs pulsed with MHC class I restricted epitopes only [103]. In that case, using specific epitopes of the TAAs tyrosinase and glycoprotein 100 (gp100), specific CD4 T cell triggering was observed, associated with enhanced vaccine efficacy. Additionally, in this case, high levels of IFNγ production and high proliferative capacity of CD4 T cells were reported upon vaccination. Encouraging findings were not only shown in melanoma, but also in breast and ovarian cancer patients. In breast cancer, the human epidermal growth factor receptor (HER-2), a protein involved in the development of this type of cancer, was targeted by DC-based vaccination [104,105]. Post-immunization sensitization of Th1 cells and increased production of IFNγ was induced, both critical features of effective antitumor immunity. In ovarian cancer, DC-based vaccines loaded with whole tumor lysates, meant to provide a wide range of antigens to reduce tumor immune escape, elicited a strong response in immunized patients [106]. The same group recently reported on increased number of CD4 and CD8 T cells and a higher survival in ovarian cancer patients treated with this type of vaccine [107]. Successful outcome was also reported using a DC vaccine loaded with mRNAs encoding for costimulatory molecules (CD70, CD40L, TLR4, called TriMix) and TAAs (gp100, MAGE-A3 or MAGE-C2), administered in combination with Ipilimumab [108]. In patients with advanced melanoma, this treatment showed a strong CD8 but a weak CD4 T cell response, probably due to the migration of the latter to the tumor site. However, an increase in the frequency of Tregs (CD3+ CD4+ CD127^low^ CD25^high^) was observed in the peripheral blood, probably induced by the production of IL-2 by CD8 T cells after vaccination. Immunization-expanded Tregs were characterized by high expression of CD62L, a molecule associated with proliferation and suppression potential in these cells. Overall, while it is recognized that this type of vaccine has multiple advantages related to safety profiles and to the potential to induce long-term effects through immunological memory [109], manufacturing concerns and induction of immune suppressive mechanisms dampening DC–T cell interactions limit the large-scale use of DC-based vaccines.

## 6. Adoptive Cell Transfer (ACT) of CD4 T Cells

Adoptive cell transfer (ACT) is a promising treatment used in immunotherapy to eradicate cancer. This strategy consists in the harvesting of T cells from a patient followed by the reinfusion after in vitro expansion to enrich in cells with antitumor properties. Further refined ACT approaches include the redirection of the T cell specificity against TAAs, by cloning TCRs and generating TCR-engineered T cells, or in the addition of chimeric antigen receptors (CARs) on the T cell surface that can recognize tumor antigens in a TCR-independent manner [110]. Generally, lymphocytes infused during ACT can derive from the cancer patient’s blood or from the solid tumor itself. Different studies reported that a successful ACT strategy relies in the infusion of CD8 cytotoxic T lymphocytes, which are also found in the neoplastic lesions, known as tumor-infiltrating lymphocytes (TILs) [111,112,113]. However, considering the importance of CD4 T cells in the TME, it is of particular interest to consider their inclusion in the ACT [114] (Figure 4). Complete regression in an HLA-DP4-positive metastatic melanoma patient treated with NY-ESO-1 specific CD4 T cells collected from the peripheral-blood was reported [115]. In this case, the authors speculated that the final outcome could be attributed to other antigens displayed by the tumor cells during their elimination, a process known as antigen spreading [116]. Results regarding safety and efficacy of the administration of peripheral blood autologous CD4 T cells were also shown using CD4 T cells engineered to express an HLA-DP4-restricted TCR targeting MAGE-A3 [117]. This regimen showed encouraging regression of tumors mediated by specific CD4 T cells in patients with metastatic cancers, even if additional studies are needed to understand the exact mechanism(s) of action. Another case report study referred about the efficacy of ACT using CD4 T cells directed against the mutated ERBB2 protein expressed by the tumor cells in a patient with a metastatic epithelial cancer, mediating a successful tumor regression [118]. In this case, a Th1 CD4 TIL polarization was observed, with cells sharing the same TCR Vbeta gene. More recently, an hTERT (human telomerase reverse transcriptase) specific TCR was isolated from a CD4 T cell clone from a vaccinated pancreatic cancer patient and expressed in primary CD4 and CD8 T cells. These cells exhibited great killing efficacy, tumor growth control and improved survival in a xenograft mouse model [119]. Interestingly, engineering of CD4 T cells with high affinity, CD8 coreceptor-independent, MHC-I restricted TCRs are very active antitumor effector cells and cooperate with redirected CD8 T cells [120]. The success of CD4 T cell ACT also arises from the use of CD4 CAR T cells, in particular in glioblastoma [121] and leukemic patients [122,123]. In these cases, the efficacy of the CD4 T cells relied on the production of IFNγ, TNF and IL-2 suggesting a predominant Th1 polarization. Moreover, CD4 CAR T cells demonstrated potent tumor eradication ability and long-term efficacy in contrast to their CD8 counterparts, which exhibited short-term effector functions, becoming rapidly exhausted upon encountering the tumor cells. Moreover, the antitumor activity of CD4 CAR T cells can be enhanced by overexpressing T-bet, a transcription factor known as a master regulator in differentiating CD4 T helper cells towards the Th1 phenotype [124]. Overall, the demonstrations of the efficacy to target TAAs using CD4 T cell transfer offers the possibility to use these cells in ACT in cancer for a personalized therapy, in combination with CD8 T cell targeting.

## 7. Cytotoxic CD4 T Cells in Immunotherapy

Besides the traditional helper and regulatory CD4 T cell classification, recent evidence demonstrated the existence of tumor-specific CD4 T cells with cytolytic capacity that can directly eliminate tumor cells [125]. Specifically, a recent work highlighted the enrichment of CD4 T cells in human bladder tumors, with a high presence of cytotoxic CD4 T cell signatures [126]. A strong correlation between tumor-specific conventional CD8 T cells and cytotoxic CD4 T cells considering the expression of GZMB, GZMK and perforin was also reported. Interestingly, the authors also showed that the cytotoxic CD4 T cells were lacking the common immune checkpoints currently targeted in immunotherapy, speculating that there should be other signals required for their regulation. Other studies previously reported about the presence of cytotoxic CD4 T cells in different human diseases [127,128,129,130], still the direct cytotoxic potential of CD4 T cells has only been shown in some cancer patients [131]. Quezada and colleagues reported the presence of cytotoxic CD4 T cells and their importance to eradicate tumors in lymphopenic mice with melanoma, particularly in association with CTLA-4 blockade [132,133]. The important aspect for tumor eradication was the differentiation in Th1 cells with the production of TNF, IL-2 and IFNγ. We recently expanded this knowledge by characterizing CD4 T cells with cytolytic capacities in melanoma patients using innovative nanobiosensors and direct sorting of TAA-specific CD4 T cells using combinatorial peptide-MHC class II multimers [18]. We reported that CD4 cytolytic T cells have a cytotoxic activity that required direct contact with the target cells and depended at least in part from granzyme B, while the co-ligation of TCR and of the molecule SLAMF7 enhanced their cytotoxicity. These findings are of particular relevance considering that different tumors are resistant to CD8-mediated rejection [134] and that some tumor cells express mainly MHC class II, while downregulate MHC class I to escape from the immune system recognition [135]. Overall, even if the origin of the cytotoxic CD4 T cells is still unclear, these findings highlight the importance of these cells in the direct killing of tumor cells. Recent observations showing that the histone deacetylases 1 and 2 (HDAC1, HDAC2) are key regulators of the differentiation of cytotoxic CD4 T cells, suggest that the use of HDAC inhibitors might be a promising strategy for the induction of these cells, by increasing their cytotoxic potential [136].

## 8. Technologies for Antigen-Specific CD4 T Cell Immune Monitoring

Different types of technologies are used to monitor the dynamics of CD4 T cells. However, these are most frequent bulk analyses performed in in vitro expanded cells, raising the necessity to develop novel methodologies to better explore the variety of antigen-reactivities present in heterogenous samples, directly ex-vivo and at the single cell level [137].

Enumeration of TAA-specific CD4 T cells is difficult, due to the low frequency of these cells in the circulation and at the tumor site. We adapted a previously established T-cell library approach [138] to estimate the frequency and isolate in an HLA-independent manner tumor-specific CD4 T cells in a high-throughput manner [139]. Though powerful, this methodology relies on the proliferative capacity of the initially seeded cells, that might be impaired in tumor-infiltrating cells. Further, isolation of tumor-specific CD4 T cells can be achieved based on the expression of activation markers, such as CD40L (CD154) [140]. Yet, only reactive cells will be detected with this strategy. Alternatively, fluorescent multimerized peptide-MHC molecules have been considered. While extensively used for the detection of CD8 T cells, even in combinatorial format [141,142], little progress has been made with pMHC class II molecules. The reasons are multiple and include the marginal role played by CD4 molecules in increasing pMHC binding avidity, the overall lower binding affinity between TCR and MHC class II complexes, the conformational diversity of pMHC complexes, the high polymorphism of MHC class II molecules and the difficulty in the generation of pMHC class II multimers [143]. In that regard, we have recently successfully developed an optimized combinatorial staining strategy using pMHC II complexes, to parallelly detect CD4 T cells of different specificities in the same sample. By using reversible multimers we increase viability of the detected antigen-specific CD4 T cells, for optimal cell culture and cloning purposes [144]. Implementation of the combinatorial format, by using DNA-barcodes or heavy metal tags in mass cytometry, is expected to increase the number of specificities detected. The main drawback of multimers remains the need of HLA genotyping and the limitation of the analysis to the HLA and antigens used to generate the reagents.

Regarding cytokine measurements, up to date, several techniques have been used including enzyme-linked immunosorbent assay (ELISA) which however does not allow real-time measurements in cell-free supernatants, or enzyme-linked immunosorbent spot (ELISpot), that relies on the use of detection/capture antibodies to enumerate cell-secreting cytokines in a cell population [145]. Drawbacks of ELISpot are that cells of interest cannot be retrieved and that the amount of secreted cytokines cannot be quantified. Multiplexed technologies also work on cell-free supernatants but employ a mixture of analyte-specific antibody precoated beads, labeled with different concentrations of a given fluorochrome, that can be visualized by flow cytometry or by other fluorescent detection systems. As alternative, cytokine production can be visualized through intracellular cytokine staining. The advantage of this technologies is the single-cell resolution measurement of secreted proteins, while a major drawback is the loss of the cells of interest due to fixation [146], and the measurement of an intracellularly produced, but not yet secreted analyte. Overall, all these assays lack spatial resolution for kinetics assessment of cytokine secretion. Microfluidics systems of arrays of subnanoliter wells (nanowells) represent an attractive system to isolate individual cells, analyze their secretome, followed by the recovery of the cells of interest. This can be achieved either using open arrays and local antibody-based capture of secreted proteins by individual cells [147], or by microengraving that consists of chips closed with a glass slide decorated with arrays of specific antibodies to capture secreted proteins. After incubation, the glass slide is removed and analytes secreted by single cells are quantified [148,149]. The disadvantage of these systems is the fluorescence-labeled imaging system used as detection mechanism, which compromises the real-time resolution of the assays. Future directions rely on the development of new biosensors that are label-free, allow real-time measurements, such as nanoplasmonic chips, since they provide high sensitivity and kinetics studies without the need of complex instrumentation [150,151].

As for cytotoxicity measurements, current state-of-the-art technologies imply the unmet need for developing techniques capable of real-time quantitative analysis at the single cell level. Microfluidics devices have been used to isolate single cells and pair effectors with targets in one-to-one ratios [152]. As easier to handle technologies, open surface microarrays combined with and automated pipeline of algorithms represent interesting alternatives [153,154,155]. We have recently developed and validated a technology relying on the use of microfabricated 2D arrays of picowells that are made of polydimethylsiloxane (PDMS) in combination with automated time-lapse fluorescence microscopy and supervised and unsupervised machine learning approaches. The microfabricated chip has a full capacity of 21,384 wells with ~65 pL in volume, allowing close interaction between a few effector and target cells in a spatially confined space [18]. With this setup, we monitored antigen-specific interactions between individual CD4 T cells and melanoma cells, used as a target. By combining multichannel time-lapse microscopy with deep neural networks, we could examine cell cytotoxicity and real-time contacts of thousands of T cells with their target in a simplified way, and at the single cell level.

## 9. Concluding Remarks

Cancer immunotherapy is experiencing a great momentum, highlighted by the 2018 Nobel Prize awarded for the discovery of immune checkpoint pathways. However, mounting challenges are currently faced to move immuno-oncology to the next-generation. Our increasing scientific knowledge on tumor-specific CD4 T cells is expected to provide new perspectives to develop optimized immunotherapy regimens. In parallel, the advancement of new technologies will offer new opportunities to study the rare and heterogenous CD4 T cell populations, to dissect the efficacy of CD4 T cell-based immunotherapies and to select the best CD4 T cell clones to optimize new immune cell products.

## Figures and Tables

**Figure 1 vaccines-09-00454-f001:**
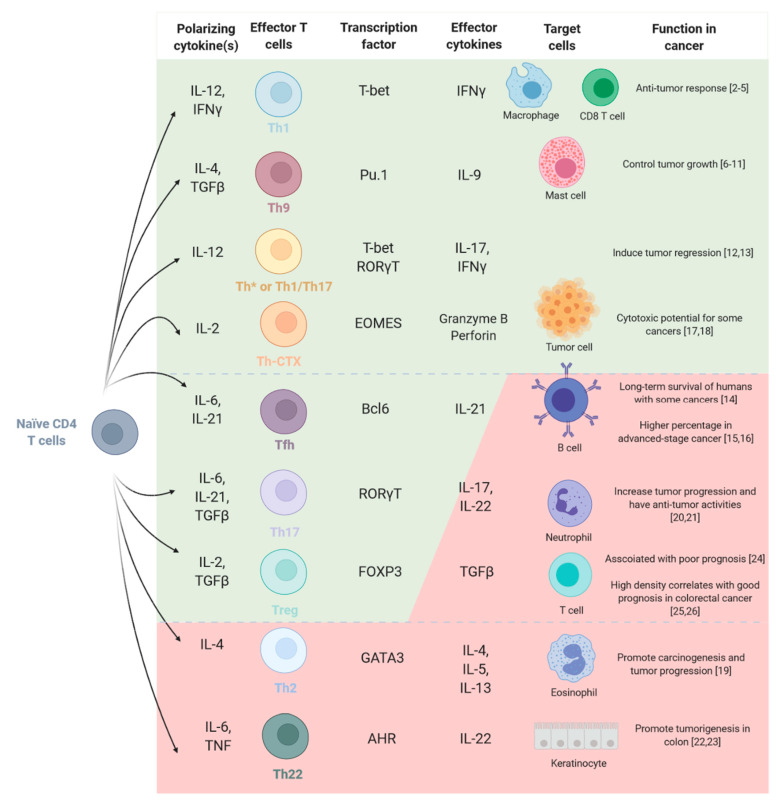
CD4 T cell subsets and tumor immunity. Each CD4 T cell polarization can be defined by the secretion of cytokines and the presence of a master transcription factor. The polarization of CD4 T cells is influenced by the TME and is linked with either tumor-promoting (red) or -suppressive functions (green).

**Figure 2 vaccines-09-00454-f002:**
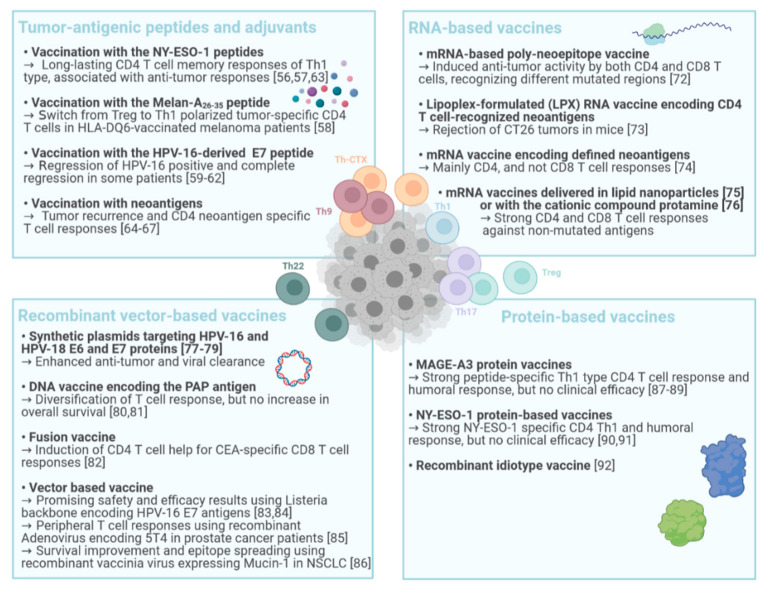
Summary of vaccination strategies and their impacts on CD4 T cell responses.

**Figure 3 vaccines-09-00454-f003:**
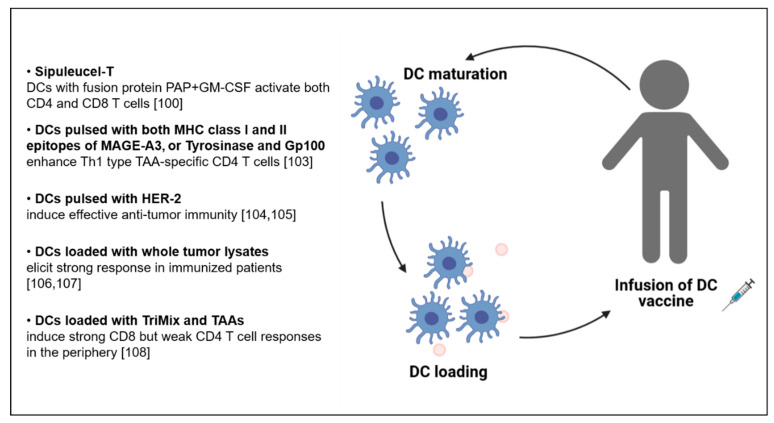
Summary of (DC)-based vaccinations for CD4 T cell activation.

**Figure 4 vaccines-09-00454-f004:**
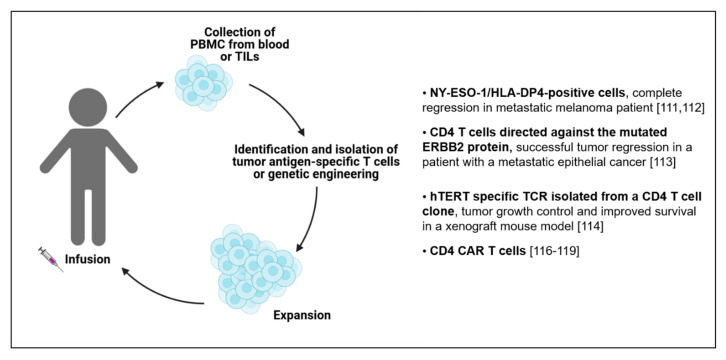
Summary of ACT therapies exploiting CD4 T cells.

**Table 1 vaccines-09-00454-t001:** Summarizing table on CD4 T cell involvement in ICB therapy.

Target	Disease (Organism)	Outcome	References
PD-1	Lung cancer (human)	70% of responding patients, dependent on pre-existing systemic CD4 T cell immunity	[33]
	Head and neck, cervical, and ovarian cancer (human)	Restoration of helper activity of exhausted PD-1^high^, CD39^+^ tumor infiltrating CD4 T cells	[34]
	Hodgkin lymphoma (human)	Clinical success associated with cytotoxicity of CD4 T cells	[35]
	Advanced gastric cancer (human)	Hyper-progression linked with expansion of tumor-infiltrating PD-1^+^ Treg	[36]
CTLA-4	Advanced melanoma (human)	Effect via the modulation of Treg cell activity and/or by the Fc portion of the antibody itself	[38]
	Pancreatic tumor (mouse)	Increased numbers of CD4 T effectors within the tumor, when ICB combined with vaccination	[41]
VISTA	*Breast cancer* (human)	Probable benefits for ongoing immunotherapy strategies	[46]
LAG-3	Metastatic melanoma (human and mouse)	Limitation of tumor growth by synergy with PD-1 blockade	[48,49]
CCR4	Melanoma (human)	Depletion of Treg and induction of tumor-antigen-specific response	[50]
OX40	Head and neck squamous cell carcinoma (human)	Increase of activated CD4 and CD8 T cells in both blood and tumor	[51]
GITR	Advanced cancers (human)	Reduction of regulatory CD4 T cells in both blood and tumor	[52]

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
