# Peer review of "Impact of Immunotherapy on CD4 T Cell Phenotypes and Function in Cancer"

_vaccines, 2021, doi:10.3390/vaccines9050454_

Round 1

Reviewer 1 Report

Jandus Camilla et al. describe in their manuscript the effects different forms of immunotherapy have on the phenotype and function of CD4+ T cells, T-Lymphocyte helper cells. CD4+ T cells can profoundly modify the tumor microenvironment and display in selected cases direct cytotoxic effects. At the end of their article they describe new techniques to measure CD4+ T cells excreted factors on a single cell level and in a reasonable throughput fashion. 

The authors review the effects the three main immunological treatment options, immune checkpoint blockade, vaccination and cell transfer, have on the role of CD4+ T cells in tumour suppression. The descriptions are detailed and supported by many references. 
Most of the effects CD4+ T cells have on the regression of tumors is via stimulating other cells of the immune system by excreted factors. 
At the end of the article the authors describe the recent efforts researchers invest to measure the secreted factors of CD4 T cells on a single cell level. 
I think the article contributes to the understanding and documentation of the complex mechanisms the immune system uses to combat the unregulated growth of tumor cells, by itself or when stimulated by therapeutic intervention.

Main subject:

This article addresses the subject of the role of CD4+ cells in fighting tumor cells after stimulation of the immune system with either immune checkpoint blockade, vaccination or stimulated eigen-cell transfer. I think the activity of the immune system upon stimulation is too variable, complex and not completely understood that a single article can possibly cover all aspects of it. Hence, the authors capture parts of it. The article refers already to 150 published research articles. I deem this to be enough. The only alternative is to reduce the breadth of the article by referring to only one method of treatment. But I, as a reader of the article, prefer that the authors cover all three methods rather than to be more exhaustive on a single one.

Considering the recent successes in cancer treatment using immune stimulation the topic is currently relevant.

The article is well structured and the introduction represents a good summary of its contents.

The review article can serve as one possible entry point to study the effects immune system stimulation may have on tumors. Because of the nature of the article being a review methodological limitations, originality and completeness of material does not really apply.

Author Response

Reviewer 1

Jandus Camilla et al. describe in their manuscript the effects different forms of immunotherapy have on the phenotype and function of CD4+ T cells, T-Lymphocyte helper cells. CD4+ T cells can profoundly modify the tumor microenvironment and display in selected cases direct cytotoxic effects. At the end of their article they describe new techniques to measure CD4+ T cells excreted factors on a single cell level and in a reasonable throughput fashion. 

The authors review the effects the three main immunological treatment options, immune checkpoint blockade, vaccination and cell transfer, have on the role of CD4+ T cells in tumour suppression. The descriptions are detailed and supported by many references. 

Most of the effects CD4+ T cells have on the regression of tumors is via stimulating other cells of the immune system by excreted factors. 

At the end of the article the authors describe the recent efforts researchers invest to measure the secreted factors of CD4 T cells on a single cell level. 

I think the article contributes to the understanding and documentation of the complex mechanisms the immune system uses to combat the unregulated growth of tumor cells, by itself or when stimulated by therapeutic intervention.

Main subject:

This article addresses the subject of the role of CD4+ cells in fighting tumor cells after stimulation of the immune system with either immune checkpoint blockade, vaccination or stimulated eigen-cell transfer. I think the activity of the immune system upon stimulation is too variable, complex and not completely understood that a single article can possibly cover all aspects of it. Hence, the authors capture parts of it. The article refers already to 150 published research articles. I deem this to be enough. The only alternative is to reduce the breadth of the article by referring to only one method of treatment. But I, as a reader of the article, prefer that the authors cover all three methods rather than to be more exhaustive on a single one. Considering the recent successes in cancer treatment using immune stimulation the topic is currently relevant. The article is well structured and the introduction represents a good summary of its contents.

The review article can serve as one possible entry point to study the effects immune system stimulation may have on tumors. Because of the nature of the article being a review methodological limitations, originality and completeness of material does not really apply.

Response to the reviewer : we thank the reviewer for his/her appreciation of our review work.

Reviewer 2 Report

The authors have discussed that how different CD4 T cell subsets can contribute to tumor immune responses during immunotherapy and the novel high-throughput immune monitoring tools that can facilitate the analysis of CD4 T cells. This well-compiled article and few minor suggestions for improvements are as follows:

  1. The novelty of the article should be clearly highlighted as number of excellent reviews have already been published on this topic
  2. The authors should also discuss about the limitations associated with the use of ICB therapy such as toxicity, high cost etc.
  3. The authors should provide their own justification and relevance of the study. This will help the readers to understand the importance of the paper.
  4. The manuscript should be carefully checked for minor typos and grammatical errors.

Author Response

Reviewer 2

The authors have discussed that how different CD4 T cell subsets can contribute to tumor immune responses during immunotherapy and the novel high-throughput immune monitoring tools that can facilitate the analysis of CD4 T cells. This well-compiled article and few minor suggestions for improvements are as follows:

  1. The novelty of the article should be clearly highlighted as number of excellent reviews have already been published on this topic.

Response to the reviewer : in this review we emphasize three aspects that have only been marginally reviewed in other review articles. The first being the need to specifically monitor antigen-specific CD4 T cell responses, as opposed to bulk CD4 T cells, to decipher immunotherapy-induced, antigen-specific versus bystander T cell activation. Second, we focus on reviewing studies that provide longitudinal monitoring of antigen-specific CD4 T cells in either pre-clinical studies of immunotherapy, or clinical trials in patients. These analyses are expected to provide critical determinants for the identification of the basis of immunotherapy efficacy. Last, we describe new tools that enable real-time and high-throughput monitoring of tumor-specific CD4 T cell functional fates down to the single cell level.

A paragraph has been added in the summary section, to highlight these aspects (page 1).

  1. The authors should also discuss about the limitations associated with the use of ICB therapy such as toxicity, high cost etc.

Response to the reviewer : we agree with the reviewer. We have added a paragraph on irAEs in ICB therapy on page 4.

  1. The authors should provide their own justification and relevance of the study. This will help the readers to understand the importance of the paper.

Response to the reviewer : Since our manuscript is a review article, rather than original research, we assume that the reviewer is referring to the rationale of this review. We have now highlighted it in our first answer to the same reviewer and at page 1, new paragraph in the summary of the revised manuscript.

  1. The manuscript should be carefully checked for minor typos and grammatical errors.

Response to the reviewer: the manuscript has been checked for typos and grammar.

Reviewer 3 Report

In this review, the authors describe the contribution of different CD4 T cell subsets to the tumor immune responses during immunotherapy. Moreover, they describe novel high-throughput immune monitoring tools that could facilitate the study of CD4 T cells, at the antigen-specific and single-cell level.

Overall, the review is well written and speaks in great depth about the role of CD4 in cancer the bibliography seems to me well updated, but in my opinion a bit difficult to follow in places. I suggest adding summary figures or tables at the end of each paragraph to help summarize the countless pieces of information at the end of CD4 T cells in immune checkpoint blockade (ICB) therapy as well as CD4 T cells in cancer vaccination, CD4 T cells in the context of dendritic cell (DC)-based vaccination, Adoptive cell transfer (ACT) of CD4 T cells and in Cytotoxic CD4 T cells in immunotherapy paragraphs.

Also for those who are not in the business, even the paragraph Cytotoxic CD4 T cells in immunotherapy are very difficult to follow, in fact, I think by itself could be another work if you want to keep in this review, I recommend to schematize the different methods with things to the advantage and or disadvantage.

Author Response

Reviewer 3

In this review, the authors describe the contribution of different CD4 T cell subsets to the tumor immune responses during immunotherapy. Moreover, they describe novel high-throughput immune monitoring tools that could facilitate the study of CD4 T cells, at the antigen-specific and single-cell level. Overall, the review is well written and speaks in great depth about the role of CD4 in cancer the bibliography seems to me well updated, but in my opinion a bit difficult to follow in places. I suggest adding summary figures or tables at the end of each paragraph to help summarize the countless pieces of information at the end of CD4 T cells in immune checkpoint blockade (ICB) therapy as well as CD4 T cells in cancer vaccination, CD4 T cells in the context of dendritic cell (DC)-based vaccination, Adoptive cell transfer (ACT) of CD4 T cells and in Cytotoxic CD4 T cells in immunotherapy paragraphs. Also for those who are not in the business, even the paragraph Cytotoxic CD4 T cells in immunotherapy are very difficult to follow, in fact, I think by itself could be another work if you want to keep in this review, I recommend to schematize the different methods with things to the advantage and or disadvantage.

Response to the reviewer: as suggested by the reviewer, we have added 1 Table and 3 schematic figures, to summarize the main information provided in the different sections.

Reviewer 4 Report

This is a review of the role of CD4 T cells and their phenotype in anti-tumor immunity by Dr. Saillard Margaux et. al.

It has been shown that there are differences in the effects of PD-1 inhibitors and anti-CTLA-4 antibodies on CD4 T cells in the mechanisms of anti-tumor immunity. This should be included in this review, along with a paper that clarifies the phenotypes of antitumor CD4 T cells based on unsupervised clustering, and how CD4 T cells and CD8 T cells work together in the presence of DC.

Important papers include,

  1. Spitzer MH, Carmi Y, Reticker-Flynn NE, et al. Systemic Immunity Is Required for Effective Cancer Immunotherapy. Cell. Jan 26 2017;168(3):487-502 e15. doi:10.1016/j.cell.2016.12.022
  2. Wei SC, Levine JH, Cogdill AP, et al. Distinct Cellular Mechanisms Underlie Anti-CTLA-4 and Anti-PD-1 Checkpoint Blockade. Cell. Sep 07 2017;170(6):1120-1133 e17. doi:10.1016/j.cell.2017.07.024
  3. Zuazo M, Arasanz H, Bocanegra A, et al. Systemic CD4 immunity: A powerful clinical biomarker for PD-L1/PD-1 immunotherapy. EMBO Mol Med. Sep 7 2020;12(9):e12706. doi:10.15252/emmm.202012706
  4. Ferris ST, Durai V, Wu R, et al. cDC1 prime and are licensed by CD4(+) T cells to induce anti-tumour immunity. Nature. Aug 2020;584(7822):624-629. doi:10.1038/s41586-020-2611-3

Unsupervised clustering by CyTOF and single cell RNAseq has led to more detailed classification of CD4 T cells. In particular, papers by Spitzer et al. and Wei et al. have used new analysis methods to identify the traits of CD4 T cell clusters that are important in tumor immunity. Anti-CTLA-4 antibodies, such as Ipilimumab, have been shown to act mainly on these CD4 T cell clusters. Therefore, it is not appropriate to review CD4 T cells with antitumor activity by ignoring these papers. However, the authors are free to change the contents in any way they chooses.

Author Response

Reviewer 4

This is a review of the role of CD4 T cells and their phenotype in anti-tumor immunity by Dr. Saillard Margaux et. al.

It has been shown that there are differences in the effects of PD-1 inhibitors and anti-CTLA-4 antibodies on CD4 T cells in the mechanisms of anti-tumor immunity. This should be included in this review, along with a paper that clarifies the phenotypes of antitumor CD4 T cells based on unsupervised clustering, and how CD4 T cells and CD8 T cells work together in the presence of DC.

Important papers include,

  1. Spitzer MH, Carmi Y, Reticker-Flynn NE, et al. Systemic Immunity Is Required for Effective Cancer Immunotherapy. Cell. Jan 26 2017;168(3):487-502 e15. doi:10.1016/j.cell.2016.12.022
  2. Wei SC, Levine JH, Cogdill AP, et al. Distinct Cellular Mechanisms Underlie Anti-CTLA-4 and Anti-PD-1 Checkpoint Blockade. Cell. Sep 07 2017;170(6):1120-1133 e17. doi:10.1016/j.cell.2017.07.024
  3. Zuazo M, Arasanz H, Bocanegra A, et al. Systemic CD4 immunity: A powerful clinical biomarker for PD-L1/PD-1 immunotherapy. EMBO Mol Med. Sep 7 2020;12(9):e12706. doi:10.15252/emmm.202012706
  4. Ferris ST, Durai V, Wu R, et al. cDC1 prime and are licensed by CD4(+) T cells to induce anti-tumour immunity. Nature. Aug 2020;584(7822):624-629. doi:10.1038/s41586-020-2611-3

Unsupervised clustering by CyTOF and single cell RNAseq has led to more detailed classification of CD4 T cells. In particular, papers by Spitzer et al. and Wei et al. have used new analysis methods to identify the traits of CD4 T cell clusters that are important in tumor immunity. Anti-CTLA-4 antibodies, such as Ipilimumab, have been shown to act mainly on these CD4 T cell clusters. Therefore, it is not appropriate to review CD4 T cells with antitumor activity by ignoring these papers. However, the authors are free to change the contents in any way they chooses.

Response to the reviewer: we agree with the reviewer. We have added and discussed the papers that he/she suggested (page 4). The work by Zuazo and colleagues was already included in our original review manuscript as number 31.

Round 2

Reviewer 4 Report

I have confirmed that this review paper has been fully revised according to my comments.